# Transcriptome Analysis Reveals Putative Induction of Floral Initiation by Old Leaves in Tea-Oil Tree (*Camellia oleifera* ‘*changlin53*’)

**DOI:** 10.3390/ijms232113021

**Published:** 2022-10-27

**Authors:** Hongyan Guo, Qiuping Zhong, Feng Tian, Xingjian Zhou, Xinjian Tan, Zhibin Luo

**Affiliations:** 1State Key Laboratory of Tree Genetics and Breeding, Key Laboratory of Silviculture of the National Forestry and Grassland Administration, Chinese Academy of Forestry, Beijing 100091, China; 2Experimental Center for Subtropical Forestry, Chinese Academy of Forestry, Xinyu 336600, China; 3Research Institute of Forestry, Chinese Academy of Forestry, Beijing 100091, China; 4College of Forestry, Central South University of Forestry and Technology, Changsha 410004, China

**Keywords:** *Camellia oleifera*, floral initiation, old leaves, transcriptome analysis

## Abstract

Floral initiation is a major phase change in the spermatophyte, where developmental programs switch from vegetative growth to reproductive growth. It is a key phase of flowering in tea-oil trees that can affect flowering time and yield, but very little is known about the molecular mechanism of floral initiation in tea-oil trees. A 12-year-old *Camellia oleifera* (cultivar ‘changlin53’) was the source of experimental materials in the current study. Scanning electron microscopy was used to identify the key stage of floral initiation, and transcriptome analysis was used to reveal the transcriptional regulatory network in old leaves involved in floral initiation. We mined 5 DEGs related to energy and 55 DEGs related to plant hormone signal transduction, and we found floral initiation induction required a high level of energy metabolism, and the phytohormones signals in the old leaves regulate floral initiation, which occurred at stage I and II. Twenty-seven rhythm-related DEGs and 107 genes associated with flowering were also identified, and the circadian rhythm interacted with photoperiod pathways to induce floral initiation. Unigene0017292 (*PSEUDO-RESPONSE REGULATOR*), Unigene0046809 (*LATE ELONGATED HYPOCOTYL*), Unigene0009932 (*GIGANTEA*), Unigene0001842 (*CONSTANS*), and Unigene0084708 (*FLOWER LOCUS T*) were the key genes in the circadian rhythm-photoperiod regulatory network. In conjunction with morphological observations and transcriptomic analysis, we concluded that the induction of floral initiation by old leaves in *C. oleifera* ‘changlin53’ mainly occurred during stages I and II, floral initiation was completed during stage III, and rhythm–photoperiod interactions may be the source of the main signals in floral initiation induced by old leaves.

## 1. Introduction

Flowering is an important feature of spermatophytes that determines whether individual plants can produce offspring successfully and affects humans by influencing crop yields [1]. Spermatophyte flowering includes three components: Floral initiation, floral development, and flowering. Floral initiation is a key step in flowering and a major phase change in spermatophytes, where developmental programs switch from vegetative growth to reproductive growth, and it can affect flowering time. Floral initiation is determined by endogenous genetic components as well as various environmental influences [2,3]. Genetic elements and molecular mechanisms of floral induction and transition have been identified in *Arabidopsis thaliana* and *Oryza sativa* during the last two decades [4,5]. Six pathways involved in plant flowering have been identified, the photoperiod pathway, vernalization pathway, gibberellic acid pathway, ambient temperature pathway, autonomous pathway, and age pathway. These regulatory pathways can work alone, but also exhibit a large amount of cross-integration. Via mutual cooperation, promotion, and inhibition between genes, different regulatory modes such as self-regulation, positive feedback regulation, and negative feedback regulation are formed, and a complex and rigorous gene regulation network ensures that plants complete the flowering process in a timely and efficient manner.

The length of the day governs a circadian rhythm, which is a crucial environmentally driven factor that controls flowering [5,6,7]. The perception of day-length information occurs in the leaves and is then transferred to the shoot apical meristem (SAM) to induce flowering. Therefore, the photoperiod pathway is closely related to the circadian rhythm. The *GI-CO-FT* module is the main photoperiod pathway in *Arabidopsis*, and *FLOWER LOCUS T* (*FT*) is a hub gene in this pathway [2]. *FLOWER LOCUS T* belongs to the phosphatidylethanolamine binding protein (PEBP) family, which can integrate signals from almost all flowering pathways. The FT protein, transferred from leaves to stem meristem through the phloem, forms a protein complex with bZIP transcription factor *FD*, which activates *SOC1*, *APETALA1* (*AP1*), and *LEAFY* (*LFY*) to induce the formation of flower meristem [8,9]. In addition to change in day length, the temperature changes with seasonal variation. In *A. thaliana*, the circadian rhythm responds to temperature via *CIRCADIAN CLOCK-ASSOCIATED 1* (*CCA1*) alternative splicing [10]. The *prr7-3 prr9-1 mutant* fails to maintain oscillation after entrainment to thermocycles and reset its clock in response to cold pulses. It is thus an important mutant that is strongly affected by temperature entrainment, indicating that *PSEUDO-RESPONSE REGULATOR7* (*PRR7*) and *PSEUDO-RESPONSE REGULATOR9* (*PRR9*) are critical components of a temperature-sensitive circadian system [11,12]. The circadian rhythm acts as an integrator of low temperatures, vernalization, and the photoperiod in bulbous plants to activate the meristem transition [13]. Therefore, the circadian rhythm is considered an important factor that affects flowering [14,15].

Plant hormones constitute an independent and cross-talk-regulated network that transforms external environmental signals and internal physiological signals, affecting plant growth and development [16]. Studies indicate that various hormones are involved in flowering regulation, including auxin, cytokinins (CKs), gibberellins (GAs), and abscisic acid (ABA) [17,18]. These studies indicate that hormone signals mainly influence flowering by affecting leaves and SAMs [19]. While hormone regulation of flowering in *A. thaliana* is likely the most studied example, it is limited to GA, but other hormones including auxin, CKs, and ABA reportedly play roles in flowering regulation [20,21]. In *A. thaliana*, GA promotes the transition from vegetative development to reproductive development and accelerates the floral transition during vernalization [22,23]. However, it plays a negative role in flower bud formation in apples and mango [24,25]. These observations indicate that hormonal regulation of flowering in perennial woody plants and the model plant *A. thaliana* differ.

To date, research on the molecular mechanisms of flowering regulation has mainly focused on a few model therophytes such as *A. thaliana* and rice. The understanding of the mechanisms involved in flowering regulation in perennials, especially woody plants, is still very limited. Although some flowering regulation pathways and action modes are relatively conserved in different plants, such as the *GI-CO-FT* component of the photoperiodic pathway and the *miRNA156 -SPLs* component of the temperature pathway, some regulation pathways differ completely between herbaceous plants and woody plants [8,22]. The tea-oil tree (*Camellia oleifera* Abel.) is an evergreen woody oil plant that belongs to the Theaceae family. Its dry seeds contain approximately 30% oil, which contains high levels of monounsaturated fatty acids and polyunsaturated fatty acids [26]. In the tea-oil tree, flowering time is a key agronomic trait that determines its yield. The flowers of the tea-oil tree are formed at the axils of new leaves of the same year, the new leaves occur in the new shoot, which develops from last year’s dormant bud, and floral initiation occurs between late March and April. In general, if floral initiation is late, blooming will be late. In the main areas of tea-oil tree production, it is usually cold and rainy during flowering time. Pollination time consists of just a few warm days, although flowering time is very long. As a result, tea-oil trees are characterized by more flowering, and less fruit and yield. Floral initiation is therefore crucial, but very little is known about the molecular mechanism of floral initiation in tea-oil trees.

## 2. Results

### 2.1. Morphological Characteristics of Floral Initiation in C. oleifera at Different Developmental Stages

The flower buds of *C. oleifera* form at the base of the vegetative buds, which grow at the terminal of new shoots or the axils of new leaves. New shoots are derived from the vegetative buds that formed the previous year, and those buds grow at the terminals of old shoots or the axils of old leaves and differentiate at the same time as the flower buds from the previous year. The species has a 1-year growth cycle. Florigenin is produced in leaves and transported by the phloem to SAM to activate floral meristem formation, which indicates that leaves play a key role in floral initiation [2,27]. The formation of flower bud primordia indicates that the floral meristem has been activated, and floral initiation has begun; floral initiation must be induced before this. Petals were the specific organ to distinguish floral buds and vegetative buds, and the differentiation of petals is considered a morphological sign of floral initiation completion in *C. oleifera* [28]. Therefore, old leaves of four key developmental stages, which are dormancy (stage I), sprouting (stage II), flower bud primordium formation (stage III), and petal differentiation (stage IV), were selected for transcriptome analysis.

The leaves formed the previous year were marked and referred to as old leaves (Figure 1a–d), and the vegetative buds in stage I (Figure 1e) grew at axils of old leaves and developed into sprouted vegetative buds (Figure 1f). At stage I, the vegetative buds grown at the axils of old leaves were dormant (Figure 1e), and only vegetative bud primordium formed (Figure 1i); no flower bud primordia formed. Old leaves at stage II were similar to those at stage I, but bud breaks occurred at this stage (Figure 1b,f), and the vegetative bud differentiated into new leaves (Figure 1j), and no flower bud primordia formed. At stage III, there was no obvious change in the morphology of old leaves, but the sprouted buds had developed into new shoots with new leaves (Figure 1g), and the flower bud primordia were formed at the base of the vegetative buds (Figure 1k). At stage IV, old leaves began to fall, and the petal primordia formed (Figure 1l).

### 2.2. Sequencing Quality Assessment

Transcriptional analyses were conducted based on RNA sequencing (RNA-seq) to elucidate the transcriptomic regulatory mechanisms of floral initiation that underlie the morphological changes in old leaves and buds at different developmental stages of *C. oleifera*. An overview of the samples for RNA-seq and the number of clean reads is presented in Table 1. The proportion of mapped reads per library ranged from 75.45% to 80.37% (Table 1). The average Q20 (sequencing error rate less than 1%) was 97.49% and the average Q30 (sequencing error rate less than 0.1%) was 92.74%. The mean GC content of all samples was 48.63%, and it ranged from 45.88% to 52.92%. De novo assembly of these clean reads with further filtering generated a total of 111,625 unigenes with a mean length of 703 bp and an N50 length of 1017 bp. A total of 1440 BUSCO groups were identified in embryophyta odb9 database, of which 1171 (81.32%) BUSCO groups were completely aligned (including 1112 single copy and 59 multiple copies), 151 were partially aligned, and 118 were not aligned. Detailed information on BUSCO analysis is presented in Appendix A. The assembly quality of the transcriptome was rated as satisfactory.

### 2.3. Functional Annotation of Unigenes

To functionally annotate the transcriptomes, similarity searches were performed for each unigene against public databases including Nr, Swiss-Prot, Clusters of Orthologous Groups of Proteins (COG/KOG), and Kyoto Encyclopedia of Genes and Genomes (KEGG). A total of 59,793 unigenes were mapped to the Nr database with an E-value threshold of 10^−5^. We identified 38,030 (34.07%), 31,494 (28.21%), and 52,785 (47.29%) unigenes that were significantly matched to entries in the Swiss-Prot, COG/KOG, and KEGG databases, respectively. A Venn diagram representing the numbers of genes annotated in these databases is shown in Appendix A. A total of 111,625 unigenes were mapped to transcriptomes of *C. oleifera* (https://github.com/Hengfu-Yin/CON_genome_data accessed on 16 August 2022) and 29,057 were matched [29]. We also detected that 31,178 and 40,013 unigenes of *C. oleifera* that exhibited significant similarity to genes found in *C. lanceoleosa* (NCBI BioProject: PRJNA780224) and *C*. *sinensis* (http://tpdb.shengxin.ren/, 7 December 2021), respectively [30].

### 2.4. Differentially Expressed Genes at Different Developmental Stages

The leaf phloem produces florigen molecules that induce a transition to reproductive development when transferred to the SAM [2]. To identify genes expressed in old leaves associated with floral initiation, six pairwise comparisons were conducted. Differentially expressed genes (DEGs) were defined using fold-changes in reads per kilobase per million reads (RPKM). DEGs were filtered with |log2 (Fold Change)| > 1 and false discovery rate (FDR) < 0.05. We paid more attention to the comparisons of two adjacent stages due to floral initiation being a developmental process. In comparison with the transcripts of genes in old leaves at stage I, the mRNA levels of 2265 genes increased markedly, and 1780 genes decreased significantly in old leaves at stage II (OLII vs. OLI, Figure 2a). Similarly, ca. expression levels of 4902 (2233 upregulated and 2669 downregulated) and 2168 (1502 upregulated and 666 downregulated) genes differed significantly in comparisons of OLIII vs. OLII and OLIV vs. OLIII, respectively (Figure 2). Approximately 1105 genes were significantly differentially expressed in both comparisons (Figure 2b). Similarly, 772 and 435 genes were significantly differentially expressed in comparisons of OLIII vs. OLII and OLIV vs. OLIII, and OLII vs. OLI and OLIV vs. OLIII, respectively (Figure 2). Approximately 162 genes were differentially expressed in all three comparisons (Figure 2). Meanwhile, the expression levels of 7024, 4902, and 4994 genes differed significantly in comparisons of OLIII vs. OLI, OLIV vs. OLI, and OLIV vs. OLII, respectively (Appendix A). Detailed information on the significant DEGs of these three comparisons is presented in Appendix A. Approximately 12 genes were differentially expressed in all six comparisons (Appendix A).

### 2.5. GO Enrichment Analysis of DEGs

To gain more information on the significant DEGs identified, a GO enrichment analysis was conducted. In that analysis, the DEGs of three pairwise comparisons were all categorized into the biological process (BP) (Figure 3), cellular component (CC) (Appendix A), and molecular function (MF) (Appendix A) categories. BP is more closely associated with floral initiation than CC and MF, so we conducted a further analysis. DEGs of three comparisons (OLII vs. OLI, OLIII vs. OLII, and OLIV vs. OLIII) enriched in the BP category were all mainly involved in four level 2 GO terms, namely metabolic processes (GO:0008152), cellular processes (GO:0009987), single-organism processes (GO:0044699), and biological regulation (GO:0065007) (Figure 3).

In the OLII vs. OLI comparison, more DEGs were upregulated than downregulated (Figure 3), and the DEGs were mainly significantly enriched in GO terms associated with phosphate-containing compound metabolic processes, and death (Figure 4a). We further identified 248 DEGs associated with death and analyzed their dynamic expression patterns. These DEGs exhibited four significant expression patterns (*p* < 0.01) (Appendix A, Profiles 1–4), 59.68% of which were upregulated at stage II. This suggested that the senescence of old leaves began at stage II. In the OLIII vs. OLII comparison, more DEGs were downregulated than upregulated (Figure 3), and DEGs were mainly significantly enriched in pathways associated with responses to localization and endogenous and external stimuli (Figure 4b). We further identified 789 DEGs associated with stimuli and analyzed their dynamic expression patterns. These DEGs exhibited five significant expression patterns (*p* < 0.01) (Appendix A, Profiles 5–9), which suggested that the old leaves responded to internal and external stimuli mainly during stage II and stage III. In the OLIV vs. OLIII comparison, more DEGs were upregulated than downregulated (Figure 3), and DEGs were mainly significantly enriched in pathways associated with defense and immunity (Figure 4c). This suggested that the old leaves were fending off external pests and diseases.

### 2.6. KEGG Enrichment Analysis of DEGs

To further identify the response mechanism of old leaves during floral initiation, a KEGG pathway enrichment analysis of the DEGs was performed. The DEGs of three comparisons were categorized into five different KEGG subsystems, including metabolism, genetic information processing, cellular processes, environmental information processing, and organismal systems (Figure 5). In OLII vs. OLI, the DEGs were significantly enriched in 21 KEGG pathways, of which secondary metabolite biosynthesis (ko01110), phenylpropanoid biosynthesis (ko00940), the metabolic pathway (ko01100), the plant hormone signal transduction pathway (ko04075), pentose and glucuronate interconversions (ko00040), sucrose metabolism (ko00500), and glutathione metabolism (ko00480) were highly significantly enriched pathways (*p* < 0.001) (Figure 6a). In OLIII vs. OLII, the DEGs were significantly enriched in 19 KEGG pathways, of which the photosynthesis-antenna proteins (ko00196), metabolic pathways (ko01100), photosynthesis (ko00195), carbon fixation in photosynthetic organisms (ko00710), biosynthesis of secondary metabolites (ko01110), flavonoid biosynthesis (ko00941), amino sugar and nucleotide sugar metabolism (ko00520), proteasome (ko03050), plant hormone signal transduction (ko04075), phenylpropanoid biosynthesis (ko00940), and carbon metabolism (ko01200) were the highly significantly enriched pathways (*p* < 0.001) (Figure 6b). In OLIII vs. OLII, the DEGs were significantly enriched in 16 KEGG pathways, of which protein processing in the endoplasmic reticulum (ko04141), flavonoid biosynthesis (ko00941), and alpha-linolenic acid metabolism (ko00592) were the highly significantly enriched pathways (*p* < 0.001) (Figure 6c).

### 2.7. DEGs Related to Energy in the Process of Floral Initiation

Essential organic nutrients accumulate to accommodate massive energy expenditure during floral initiation. Five genes, Unigene0000880, Unigene0025066, Unigene0017865, Unigene0011921, and Unigene0087677, all associated with starch and sucrose metabolism, exhibited higher expression levels at stages I and II (Figure 7). Of all the genes involved in sucrose and starch metabolism during floral initiation in *C. oleifera*, these five genes exhibited the highest expression levels and the most significant changes. These findings indicate that high levels of energy metabolism occurred in old leaves at stages I and II, and also implied that floral initiation induced by old leaves mainly occurred at stages I and II.

### 2.8. DEGs Related to Plant Hormone and Signal Transduction

It has been reported that endogenous hormones can advance or delay the floral transition, which has important effects on flowering times and crop yields. A total of 55 DEGs were identified after merging the DEGs yielded by three comparisons (OLII vs. OLI, OLIII vs. OLII, and OLIV vs. OLIII). These DEGs were involved in several plant hormone signal transduction pathways, including 19 DEGs in the auxin signaling pathway, 11 in the cytokinin (CK) signaling pathway, 9 in the gibberellins (GAs) signaling pathway, and 16 in the abscisic acid (ABA) signaling pathway (Appendix A). Unigene0101040, Unigene0007576, and Unigene0093625 were associated with the highest expression levels of auxin, CKs, and GAs, respectively, and exhibited similar expression patterns, with significantly higher expression levels at stage II. Unigene0102339 was associated with the highest expression level of ABA, and exhibited significantly higher expression levels at stages I, II, and IV.

### 2.9. DEGs Related to Rhythm in the Process of Floral Initiation

Based on gene expression and results of GO and KEGG enrichment analysis, 27 rhythm-related DEGs were identified, and their expression levels are shown in Figure 8. The majority were related to the photoperiodic response and exhibited significantly higher expression levels at stages I and II. Expression levels of Unigene0046809 (*LHY*) and Unigene0017292 (*PRR*), which were the core genes of the circadian rhythm, were upregulated more significantly. Expression levels of Unigene0084708 (*FT*) exhibited similar patterns to those of *LHY* and *PRR*.

### 2.10. Identification of Flowering-Related Genes in C. oleifera

Based on the annotation of unigenes, genes that were expressed at a maximum RPKM <1 in all samples were removed. A total of 107 genes associated with flowering were obtained and are shown in Appendix A. These included photoperiod pathway genes such as Unigene0011463 (*FKF*), Unigene0009932 (*GI*), Unigene0006307 (*CDF*), and Unigene0001842 (*CO*); vernalization pathway genes including Unigene0099508 (*VRN1*); gibberellic acid pathway genes including Unigene0093626 (*GAI*), Unigene0007406 (*GAI*), Unigene0093623 (*GAI*), Unigene0093624 (*GAI*), and Unigene0093625 (*GAI*); ambient temperature pathway genes including Unigene0098370 (*SVP1*) and Unigene0040069 (*SVP2*); autonomous pathway genes including Unigene0052335 (*FY*), Unigene0091286 (*FPA*), and Unigene0119870 (*FCA*); and age pathway genes including 27 belonging to the squamosa promoter-binding protein family. In addition, the key integrator of the flowering regulation signal Unigene0084708 (*FT*) was obtained, which exhibited high expression at stage II. This suggested that stage II may be the key stage for the induction of flowering by old leaves. All unigenes were important resources for the study of flower initiation and flower organ formation in the future.

### 2.11. Quantitative Real-Time PCR Analysis

Quantitative real-time PCR (qRT-PCR) analysis was used to confirm the accuracy of the RNA-seq data. Twelve DEGs related to flowering, rhythm, and hormones were chosen for qRT-PCR analysis of all samples. The variation patterns of DEG expression levels identified via qRT-PCR were consistent with the RNA-seq data (Figure 9 and Appendix A). This indicated that the RNA-seq data obtained in the study were reliable.

## 3. Discussion

The timing of reproduction is an important component of adaptation strategies in plants, and flowering is a key element of reproductive growth in higher plants. Flowering is also an important factor affecting the fruits and seed yields of many agricultural and forestry crops. The transition to flowering in plants is triggered by a number of integrated seasonal signals including photoperiod, temperature, age, and humidity [3,31]. However, no previous research in this area has been conducted on *C. oleifera*. Thus, it is necessary to investigate relationships between floral initiation and the key genes that regulate the process.

### 3.1. Plant Hormone and Signal Transduction during Floral Initiation

There is extensive cross-talk between different hormonal pathways involved in the regulation of floral initiation, which makes plants more flexible with respect to responding to environmental signals. Hormonal signaling cascades affect the transcription of floral integrators that act in the leaf or SAM [4,16]. Four types of phytohormones—auxin, CKs, GAs, and ABA—are evidently associated with flowering, although their functions and the molecular mechanisms involved in their regulation of floral transition are not particularly clear [32,33]. Gibberellins are one of the most studied phytohormones and are considered to be the most important plant hormones in the regulation of floral transition. In *A. thaliana*, GAs promote flowering by inducing *SOC1*, *FT*, *TSF*, *SPL*, and *LFY* expression. *GIBBERELLIN INSENSITIVE DWARF1* (*GID1*) is reportedly a soluble GA receptor [34,35]. After *GID1* was activated by GA signaling, it specifically bound to DELLA proteins (DELLAs) to promote their proteasomal degradation. The GID1-GA-DELLA complex stabilizes receptor–hormone interactions. The GID1-GA complex regulates plant growth and development by downregulating DELLA repressors via direct protein–protein interaction [36]. DELLAs contain conserved DELLA and VHYNP sequences in the N-terminal regulatory region, which are important for GID1 binding and proteolysis of the DELLA proteins, and the conserved GRAS domain in the carboxy-terminal half. Five DELLA genes have been reported in *A. thaliana*, namely *REPRESSOR OF GA1-3* (*RGA*), *GA INSENSITIVE* (*GAI*), *RGA-LIKE 1* (*RGL1*), *RGL2*, and *RGL3* [37]. In SD, GA-mediated DELLA degradation can promote flowering independently of *FT*, *TSF*, and *PHYTOCHROME INTERACTING PROTEINs* (*PIFs*) [38]. In the current study, one differentially expressed *GID1* gene (Unigene0018743) exhibited lower expression at stage II. Four DELLA genes (Unigene0007406, Unigene0093624, Unigene0093625, Unigene0093624) exhibited higher expression at stage II. The consistency of *GID1* and DELLA gene expression patterns indicated that GA signaling promoted *C. oleifera* flowering initiation at stage II, and *GID1* and *DELLA* may be the most important genes involved in the regulation of GA signal transduction during *C. oleifera* flowering initiation.

Gibberellins is not the only hormone that regulates flowering transition. Other hormones can act together with GA to control the flowering transition. The auxin signaling component INDOLE-3-ACETIC ACID 7 (IAA7)/AUXIN RESISTANT 2 (AXR2) is likely to be involved in the suppression of floral transition under SDs in a GA-dependent manner [39]. The AUXIN (AUX)/INDOLE-3–ACETIC ACID (IAA) family is generally thought to act as a repressor of auxin-induced gene expression [40]. Auxin binding to the auxin receptor (TIR1/AFB family of F-box proteins) triggers the degradation of AUX/IAAs. In the absence of auxin, these AUX/IAAs physically interact with and inhibit the activity of transcriptional regulators AUXIN-RESPONSE FACTORs (ARFs) [41]. In the present study, two TIR1 (Unigene0051840, Unigene0051841), three ARF (Unigene0064360, Unigene0008851, Unigene0091572), and five IAA genes (Unigene0046669, Unigene0118723, Unigene0046669, Unigene0097461, Unigene0101040) were identified as DEGs in three comparisons. The expression patterns of TIR1 and IAA genes were highly similar to those of *GID1* and *DELLA* genes, which were the key genes in the gibberellin signal transduction pathway. Thus, the relationship between gibberellin and auxin in the regulation of flowering initiation in *C. oleifera* requires further investigation.

Cytokinins are strongly associated with the early events of flowering initiation and induce flowering by activating *TSF* in *A. thaliana* [42,43]. Histidine kinases (HKs), histidine phosphotransfer proteins (HPs), and response regulators (RRs) constitute the CK signal transduction pathway [44]. In *A. thaliana*, the CK signal is received by three membrane receptors called ARABIDOPSIS HK 2 (AHK2), AHK3, and AHK4/CRE1, whereby *A. thaliana* HPs (AHPs) mediate His-to-Asp phosphotransfer from the cytoplasm to the nucleus. In the current study, three AHKs (Unigene0000956, Unigene0001841, and Unigene0000956) and three AHPs (Unigene0007576, Unigene0051063, and Unigene0092870) were among the DEGs identified in the three comparisons. Expression levels of AHKs and AHPs in OLI and OLII were higher than those in OLIII and OLIV, indicating that old leaves received more CK signals at stages I and II. Type-B ARR (B-ARR) transcription factors then directly modulate the expression of type-A ARR (A-ARR) primary CK target genes, while A-ARR functions to negatively regulate B-ARR expression, as a negative feedback loop that controls CK responses [45,46]. In the present study, one B-ARR gene (Unigene0017776) and four A-ARR genes (Unigene0033968, Unigene0051730, Unigene0081652, and Unigene0091200) were among the DEGs identified in the three comparisons. The expression pattern of A-ARRs was similar to that of B-ARR, and all of them exhibited higher expression levels in OLI and OLII. CK involvement has been reported in the regulation of flowering in many plants, including rose [45], longan [47], apple [48], and *Fragaria vesca* [49]. The differential expression patterns of AHKs, AHPs, B-ARR, and A-ARRs in the initiation of flowering indicate that CKs are involved in flowering initiation in *C. oleifera* during stages I and II.

Abscisic acid involvement in plant flowering regulation is well established, but the contribution of ABA signaling in the flowering transition remains controversial [16,50,51]. ABA is considered to be a repressor of the transition to flowering in plants because it negatively regulates it by directly promoting FLC transcription in *A. thaliana* [52]. Endogenous ABA upregulates FT expression to positively regulate flowering under long-day conditions [53,54]. PYRABACTIN RESISTANCE (PYR), PROTEIN PHOSPHATASE 2Cs (PP2Cs), and SNF1-RELATED PROTEIN KINASE 2s (SnRK2s) are three signaling components that constitute the core ABA signaling pathway [55]. In the present study, five PYR (Unigene0000391, Unigene0025169, Unigene0055998, Unigene0080065, and Unigene0102339), two PP2C (Unigene0071501 and Unigene0067823), and six SnRK2 (Unigene0050823, Unigene0075471, Unigene0067963, Unigene0101500, Unigene0103492, and Unigene0118131) genes were among the DEGs identified in the three comparisons. Unigene0000391 and Unigene0102339 were the most important genes in the ABA signaling pathway during flowering initiation at the expression level. Their expression patterns differed from those of *GID1* and DELLAs. This indicated that different signal transduction modes of ABA and GA occurred during the initiation of flowering in *C. oleifera*. ABA and GA antagonistically mediate diverse plant flowering, and an optimal balance between ABA and GA is essential for plant growth and development [56]. The expression levels of Unigene0000391 and Unigene0102339 were significantly lower during OLII and OLIII, respectively. *GID1* and DELLA genes exhibited higher expression during OLII, implying that ABA and GA may play antagonistic roles in the initiation of flowering in *C. oleifera*. These results suggest that plant hormones may regulate flowering initiation in *C. oleifera* at stages I and II, and that the process may involve complex interaction, but the regulatory mechanisms involved require further investigation.

### 3.2. Responses to Circadian Rhythms in Old Leaves during Flowering Initiation

Plants monitor photoperiod changes via leaves to synchronize their flowering with seasonal changes and maximize reproduction [57]. The circadian rhythm is intricately connected to the photoperiodic pathway. A fundamental role of the circadian rhythm is to predict daily and seasonal environmental cycles, which allows plants to optimize internal processes based on external conditions [15]. Another role of the circadian rhythm is to measure the length of the daily light period in leaves, and convey a signal to the shoot apex to initiate floral initiation accordingly [58]. In *A. thaliana*, the circadian oscillator at the core of this system is composed of the interlocked feedback loop formed by the major transcription factors *CIRCADIAN CLOCK-ASSOCIATED 1* (*CCA1*), *LATE ELONGATED HYPOCOTYL* (*LHY*), *TIMING OF CAB EXPRESSION 1* (*TOC1*), and *PSEUDO-RESPONSE REGULATOR* (*PRR*). The morning-expressed Myb transcription factors CCA1 and LHY suppress evening-phase clock genes such as *TOC1*, *PRR5*, *LUX ARRHYTHMO* (*LUX*), *EARLY FLOWERING 3* (*ELF3*), *EARLY FLOWERING 4* (*ELF4*), and *GI* via their recruitment of *DEETIOLATED1* (*DET1*) [59]. *CCA1* and *LHY* activate the expression of *PSEUDO-RESPONSE REGULATOR7* (*PRR7*) and *PSEUDO-RESPONSE REGULATOR9* (*PRR9*) in the morning, then *PRR7* and *PRR9* repress the transcription of *CCA1* and *LHY* during the rest of the day. In contrast, evening-expressed *TOC1* activates *CCA1* and *LHY* expression. The expression of *PHYTOCHROME B (PHYB)* decreased gradually in deciduous leaves. In *C. oleifera*, old leaves were the most important receivers of environmental signals during flowering initiation. In the present study, 27 rhythm-related DEGs were identified. *LHY* and *PSEUDO-RESPONSE REGULATOR5* (*PRR5*) were the two genes with the greatest changes in expression. They were highly expressed during stages I and II and exhibited similar expression patterns. *LHY* expression at stage I was 90 times that at stage III, and 42 times that at stage IV. *LHY* expression at stage II was 137 times that at stage III and 65 times that at stage IV. The expression of these two genes was synchronized with *FT* during stages I and II. These observations suggested that the responses of old leaves to the circadian rhythm mainly occurred during stages I and II, and that *LHY* and *PRR5* may be key genes in the responses to the circadian rhythm during flowering initiation in *C. oleifera*.

### 3.3. Putative Gene Regulatory Network of Floral Initiation in C. oleifera

Flowering during appropriate conditions can increase the success of reproduction and crop yields, so plants have evolved a set of genetic and metabolic mechanisms to sense and adapt to their changing environment. The mechanisms of flowering in the model plant *A. thaliana* have been well studied in recent decades, and a complex network has been proposed [60]. Day length is a major environmental factor that regulates the initiation of flowering. Although longer day lengths promote flowering while shorter day lengths inhibit flowering in *A. thaliana*, the flowering time of soybean under short-day conditions was earlier than that under long-day conditions [61]. This indicates that the response mechanism of flowering to the photoperiod is different in diverse plants, and the response to the photoperiod is also different between woody and herbaceous perennials [24,62]. In annual herbaceous plants, we focus on the transition from vegetative to reproductive growth, while in woody plants harvested for fruit, more attention is paid to flowering during the annual growth cycle of adult trees. Even in woody species, the flowering mechanism is different between young and adult trees, where the flowering mechanism is determined in the immature bud in young trees, whereas it is determined in the leaves in adult trees [63]. This is the main reason for selecting old leaves as materials. It has been reported that longer day lengths promote earlier flowering of *C. oleifera*, and flavonoid biosynthesis, amino sugar and nucleotide sugar metabolism, and circadian rhythm pathways may function in the photoperiodic flowering pathway of *C. oleifera* [64]. It takes approximately 10 months from floral initiation to flowering in *C. oleifera*, but whether the photoperiod promotes early flowering by promoting floral initiation or floral organ development is uncertain. The results of the current study provide insight into the regulatory relations of floral initiation in oil-tea plants, in which the circadian rhythm and photoperiod may function in floral initiation (Figure 10). The results of this study are similar to those previously reported and further confirm the important role of the photoperiod in the flowering of *C. oleifera* [64].

The leaves of plants sense environmental signals such as photoperiods and temperatures, and this information converges in *FT* and is transported to the SAM to regulate flowering initiation. Plant leaves are the most important organ with respect to receiving information on day length and the circadian rhythm. *PRRs*, *CCA1*, *LHY*, *TOC1*, and *GI* are components of the circadian clock, and Unigene0046809 (LHY) and Unigene0017292 (PRR) expression are associated with flowering initiation in *C. oleifera*. The transcriptional expression of *GI, CO*, and *FT* in the photoperiodic pathway is strictly regulated by the circadian rhythm [65], and their expression levels were synchronized with Unigene0046809 (LHY) and Unigene0017292 (PRR) in the current study. This indicated that the photoperiodic pathway is regulated by the circadian rhythm to promote flowering initiation in *C. oleifera*. Light information is also received by photoperiod genes such as *FLAVIN-BINDING*, *KELCH REPEAT*, *F-BOX 1* (*KFK1*), *CYCLING DOF FACTORs* (*CDFs*), and *CO*. Under long-day conditions, light can promote the binding of *GI* to FKF1 protein, and the GI-FKF1 complex is formed in a blue-light-dependent manner and mediates the degradation of CO transcriptional repressors known as CYCLING DOF FACTORs (CDFs). *CO* transcriptional inhibition can promote flowering initiation via the activation of *FT* [65]. In the present study *FKF*, *GI*, and *CO* were highly expressed during stage I, and *FT* was expressed during stages I and II. Expression peaked during stage II and then decreased to undetectable levels, indicating that photoperiod induction of flowering initiation began during stage I, peaked during stage II, and was completed during stage III. The appearance of flower bud primordia also indicated the completion of flowering initiation (Figure 1).

## 4. Materials and Methods

### 4.1. Plant Growth and Sample Collection

The 12-year-old *C. oleifera* (cultivar ‘changlin53’) grew well without disease and with similar growth potential, which was planted under natural conditions in an experimental field of the Experimental Center for Subtropical Forestry, Chinese Academy of Forestry (ECSF, CAF), located in Fenyi County, Jiangxi Province, China (27°49′ N, 114°39′ E; altitude 88–92 m). The same fertilizer and water management were conducted for plants. From 12 February to 4 June 2019, bud and old leaves samples were collected every 7d for morphological analysis. According to the morphological results [28], four key developmental stages were selected in the current study: Dormancy (stage I), sprouting (stage II), flower bud primordium formation (stage III), and petal differentiation (stage IV). After sampling, the tissues were snap frozen in liquid nitrogen and stored at −80 °C prior to RNA isolation. Three biological replicates were used for each of the sampling points, and each biological replicate involved three mixed plant samples.

### 4.2. Analysis of Morphological Characteristics

Before each sampling, the experimental materials were photographed with a camera and a stereoscopic microscope. Buds were stored in formaldehyde-acetic acid-ethanol fixative (FAA) stationary liquid with 5% formaldehyde, 5% acetic acid, 63% anhydrous ethanol, and 27% pure water prior to use. The samples were dehydrated in ethanol concentrations of 70% (two times, 30 min each time), 80% (one time, for 30 min), 90% (one time, for 30 min), 95% (one time, for 30 min), and 100% (two times, 30 min each time). The dehydrated samples were then dried via the critical point drying method using a liquid CO_2_ device (Model HCP-2, Hitachi, Tokyo, Japan), and gold-coated using an Edwards E-1010 ion sputter coater (Hitachi, Tokyo, Japan). The prepared samples were observed with an S4800 variable pressure scanning electron microscope (Hitachi), and all images were collected for further analysis [66].

### 4.3. Transcriptome Sequencing and Functional Annotation of the Transcriptome

Total RNA was extracted from all samples using the RNA prep Pure DP441 (Tiangen, Beijing, China) in accordance with the manufacturer’s protocol. Integrity and pollution were measured via agarose gel electrophoresis, RNA purity was measured with a NanoDrop Micro-spectrophotometer, RNA concentration accuracy was measured using a Qubit 2.0 Fluorometer, and accurate integrity was measured using an Agilent 2100 bioanalyzer (Agilent Technologies, Santa Clara, CA, USA). Twelve RNA samples were used for library preparation and RNA sequencing by Gene Denovo Biotechnology Co., Guangzhou, China. Sequencing data quality control includes sequencing data statistics, original data statistics, and quality control data statistics. Transcriptome denovo assembly was carried out with short reads assembling program Trinity (Version v2.8.4, kmer_size set to 31, min_kmer_cov set to 17, normalize_max_read_cov set to 50, other parameters set to default) [67]. Trinity is a modular method and software package, which combines three components: *Inchworm*, *Chrysalis*, and *Butterfly*. Firstly, *Inchworm* assembles reads by a greedy k-mer-based approach, resulting in a collection of linear contigs. Next, *Chrysalis* clusters related contigs that correspond to portions of alternatively spliced transcripts or otherwise unique portions of paralogous genes, and then builds a de Bruijn graph for each cluster of related contigs. Finally, *Butterfly* analyzes the paths taken by reads and read pairings in the context of the corresponding de Bruijn graph, and outputs one linear sequence for each alternatively spliced isoform and transcripts derived from paralogous genes [67]. The longest transcript in gene level 3 classification of Trinity assembly results was selected as unigene. Busco was used to assess assembly quality. The unigene expression was calculated and normalized to RPKM [68]. The resources used to annotate gene function were the Nr (NCBI non-redundant protein sequences), Swiss-Prot (a database of manually annotated and reviewed protein sequences), Protein Family (Pfam), COG/KOG, KEGG, and Gene Ontology (GO) databases.

### 4.4. Differential Expression Genes Analyses

The DEseq2R package (v1.20.0, parameters set to default) was used to conduct pairwise comparisons. Read counts for each gene were normalized to RPKM, which is currently the most widely used method for estimating gene expression levels and representing gene expression levels. There are strict algorithms for screening DEGs in two samples. Fold changes in differential gene expression across six pairwise comparisons were calculated based on RPKM values. Multiple hypotheses using FDR tests were conducted between two samples, and the corrected FDR value < 0.05 and |fold change| > 1 were used as the criteria to determine significant differences in gene expression. To identify genes expressed in old leaves associated with floral initiation, six pairwise comparisons were conducted, which were OLII vs. OLI, OLIII vs. OLI, OLIV vs. OLI, OLIII vs. OLII, OLIV vs. OLII, and OLIV vs. OLIII.

### 4.5. GO and KEGG Enrichment Analysis of DEGs

KEGG and GO enrichment analyses were conducted to investigate pathways and biological functions that were activated during flower transition in *C. oleifera*. The background documents of GO and KEGG were from GO and KEGG annotation. GO and KEGG enrichment analyses of DEGs were performed using OmicShare tools in 2021, a free online platform for data analysis (http://www.omicshare.com/tools, 16 August 2022).

### 4.6. qRT-PCR Analysis

Total RNA was extracted from the same plant used for RNA sequencing. cDNA was synthesized using a PrimeScript^TM^ RT reagent kit with gDNA Eraser (Perfect Real Time) (TaKaRa), in accordance with the manufacturer’s protocol [69]. cDNA reverse-transcription products were used as templates for qRT-PCR. qRT-PCR was performed using TB Green^®^ Premix Ex Taq™ II (Tli RNaseH Plus) (TaKaRa), in accordance with the manufacturer’s instructions. Relative expression of the target gene in each sample was calculated via the 2^−ΔΔCt^ method [69]. Relative expression values derived from qRT-PCR and RPKM values of RNA-seq were compared. *CoGAPDH* and *CoTUB1* were used as internal control. All primers for qRT-PCR are listed in Appendix A.

## 5. Conclusions

The current study comprehensively compared transcriptomes from old leaves at different developmental stages during flowering initiation in *C. oleifera* ‘changlin53’. Five DEGs related to energy and 55 related to plant hormone signal transduction were identified. The induction of flowering evidently required a high level of energy metabolism, and phytohormone signals in old leaves regulated flowering initiation during stages I and II. Twenty-seven rhythm-related DEGs and 107 genes associated with flowering were identified, and analyses indicated that the circadian rhythm interacted with photoperiod pathways to induce flowering initiation. Unigene0017292 (PRR), Unigene0046809 (*LHY*), Unigene0009932 (*GI*), Unigene0001842 (*CO*), and Unigene0084708 (*FT*) were key genes in the circadian rhythm–photoperiod regulatory network. In conjunction with morphological observations and transcriptomic analysis, we concluded that the induction of flowering initiation by old leaves in *C. oleifera* ‘changlin53’ mainly occurred during stages I and II and was completed during stage III, and that the rhythm-photoperiod may be the main signal in flowering initiation induced by old leaves.

## Figures and Tables

**Figure 1 ijms-23-13021-f001:**
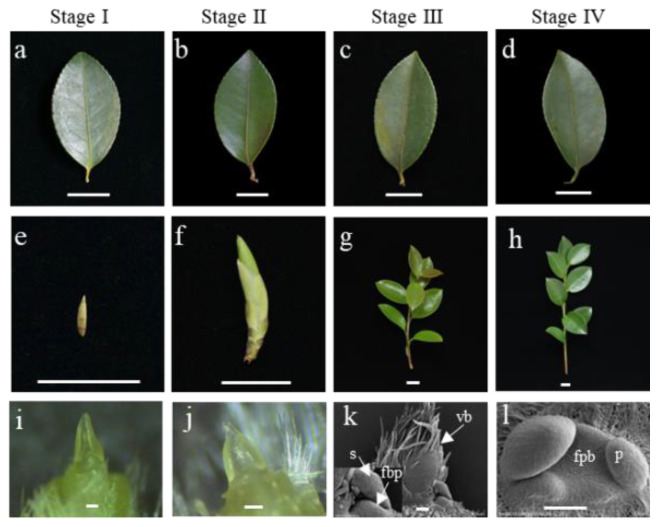
Developmental stages of old in *C. oleifera*. Old leaves (**a**–**d**) are referred to as the leaves formed in the last year. Vegetative buds in stages I (**e**) are grown at axils of old leaves and will develop into sprouted buds (**f**) and differentiate into new leaves (**i**,**j**). The new shoots in stages III (**g**) and IV (**h**) are derived from sprouted buds in panel (**f**). Flower buds in stages III and IV were located at the base of the vegetative buds of new shoots (**k**,**l**). vb: Vegetative bud, fpb: Flower primordium bud, s: Sepal, p: Petal. The scale bars in (**a**–**h**) and (**i**–**l**) indicate 1.5 cm and 100 μm, respectively.

**Figure 2 ijms-23-13021-f002:**
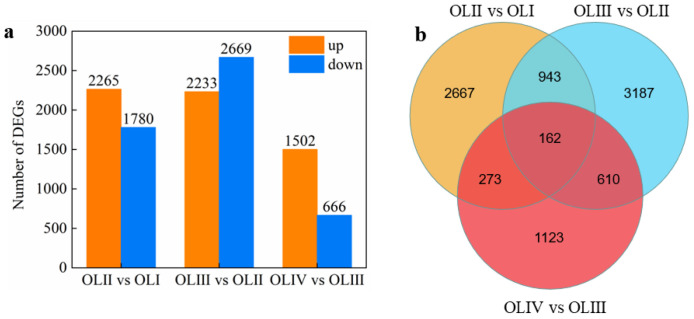
Numbers of DEGs in diverse pairwise comparisons (**a**) and Venn diagram showing the common and unique DEGs among different pairwise comparisons (**b**). Detailed information on the significant DEGs is presented in Appendix A.

**Figure 3 ijms-23-13021-f003:**
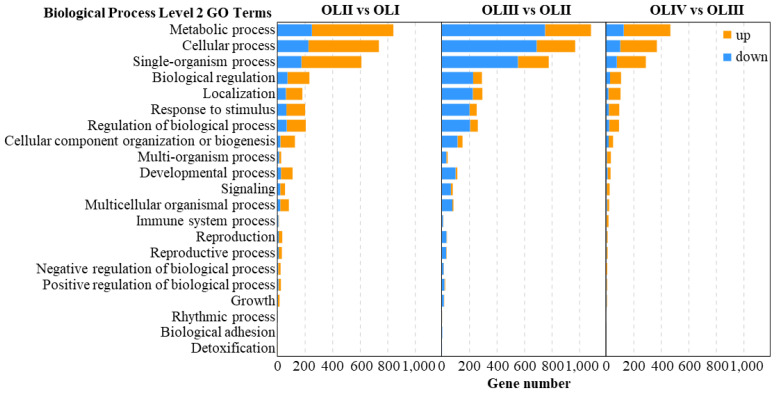
GO functional of the significantly differentially expressed genes of old leaves in diverse pairwise comparisons.

**Figure 4 ijms-23-13021-f004:**
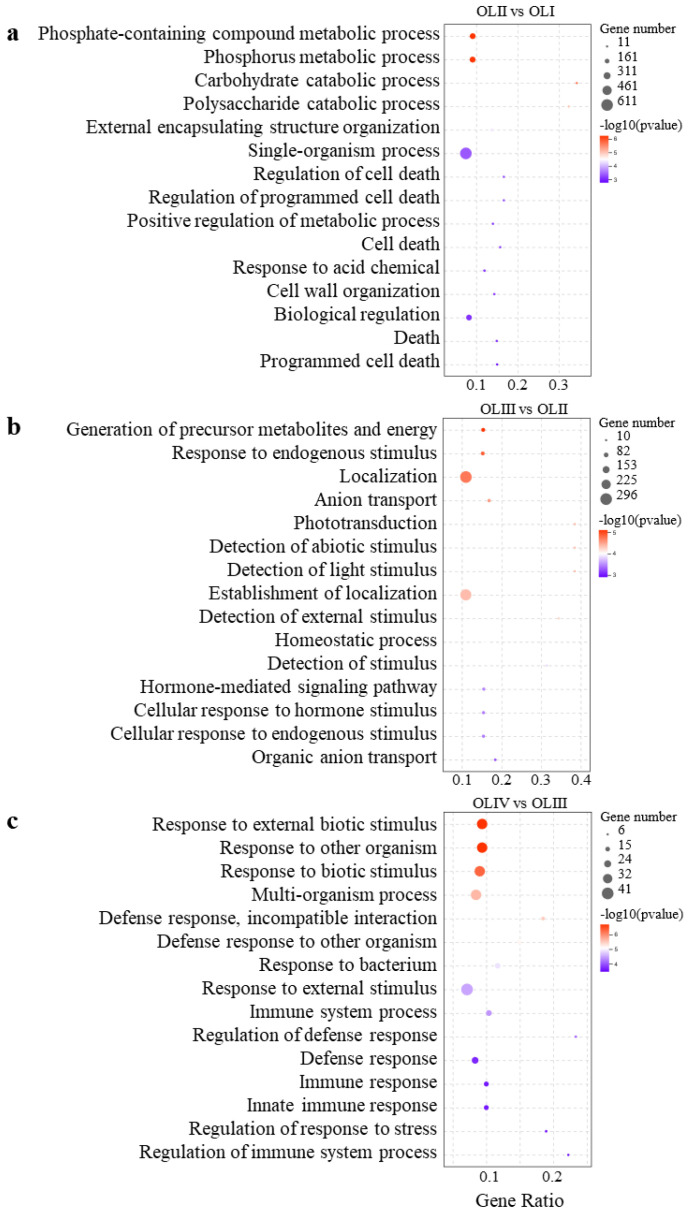
Top 15 significance level GO terms in OLII vs. OLI (**a**), OLIII vs. OLII (**b**) and OLIV vs. OLIII (**c**) comparisons.

**Figure 5 ijms-23-13021-f005:**
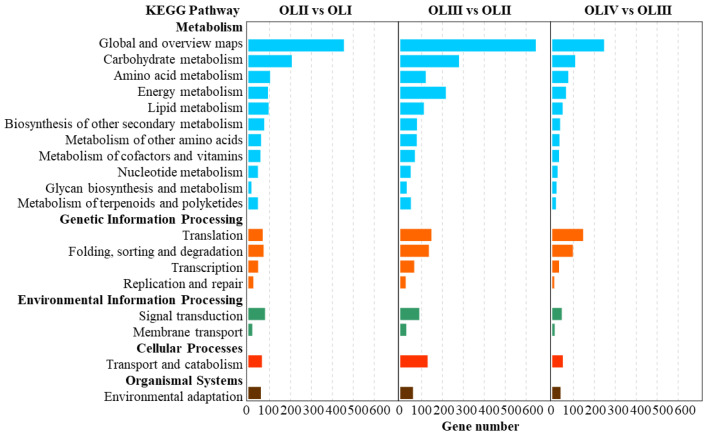
KEGG pathway enrichment analysis of the significantly differentially expressed genes of old leaves in diverse pairwise comparisons.

**Figure 6 ijms-23-13021-f006:**
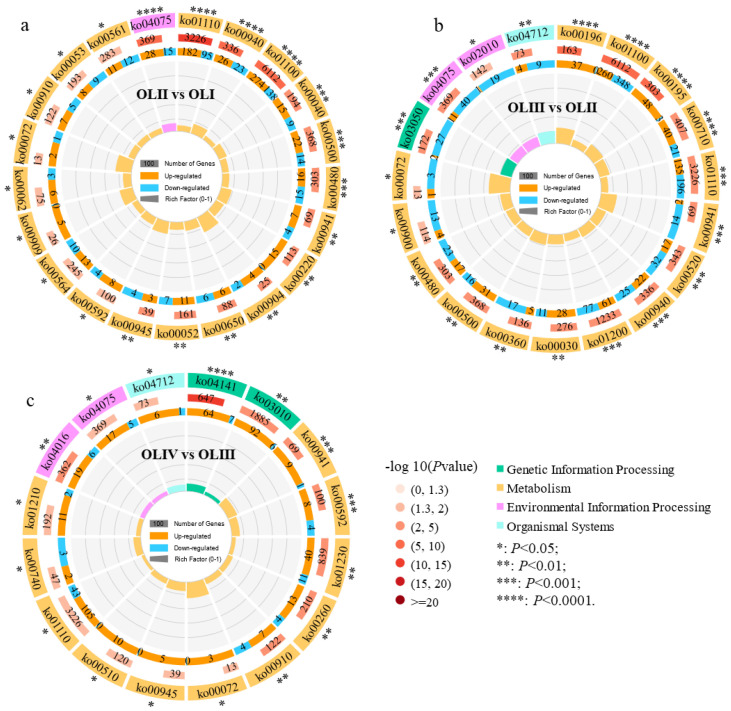
Significantly enriched KEGG pathway in diverse pairwise comparisons (**a**–**c**) (*p* value < 0.05).

**Figure 7 ijms-23-13021-f007:**
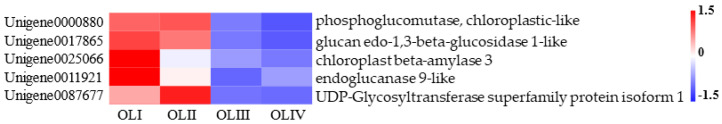
The heatmap showed the DEGs associated with starch and sucrose metabolism.

**Figure 8 ijms-23-13021-f008:**
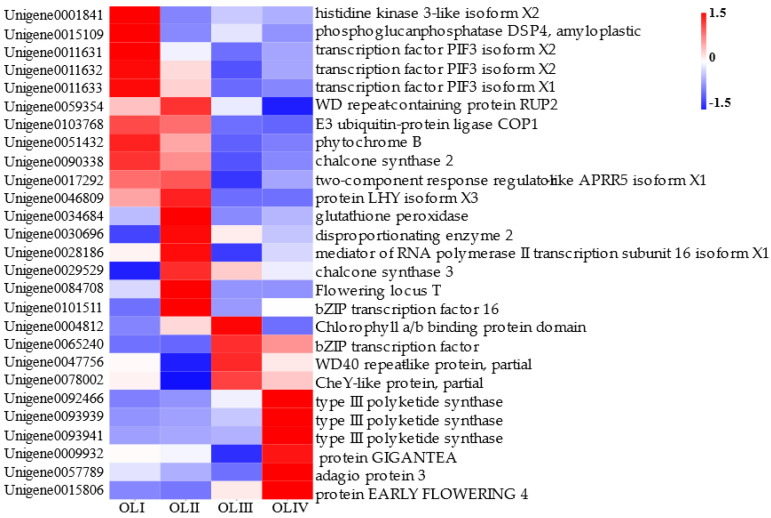
The heatmap showed the DEGs related to rhythm.

**Figure 9 ijms-23-13021-f009:**
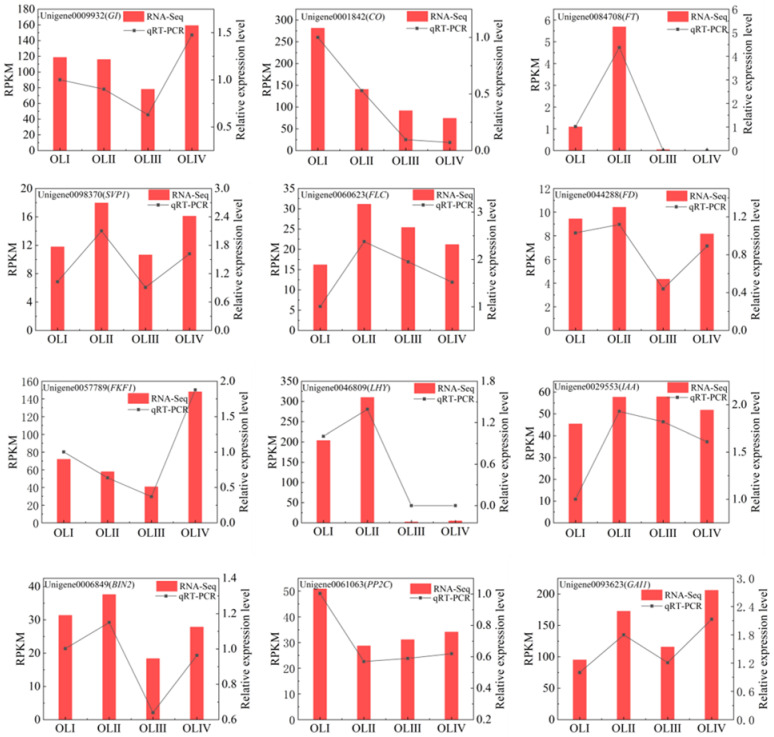
The comparison of the expression levels of 12 DEGs identified related to flowering, rhythm, and hormone between RNA-Seq and qRT-PCR analyses from old leaves at different developmental stages during floral transition. *CoTUB1* is used as internal control.

**Figure 10 ijms-23-13021-f010:**
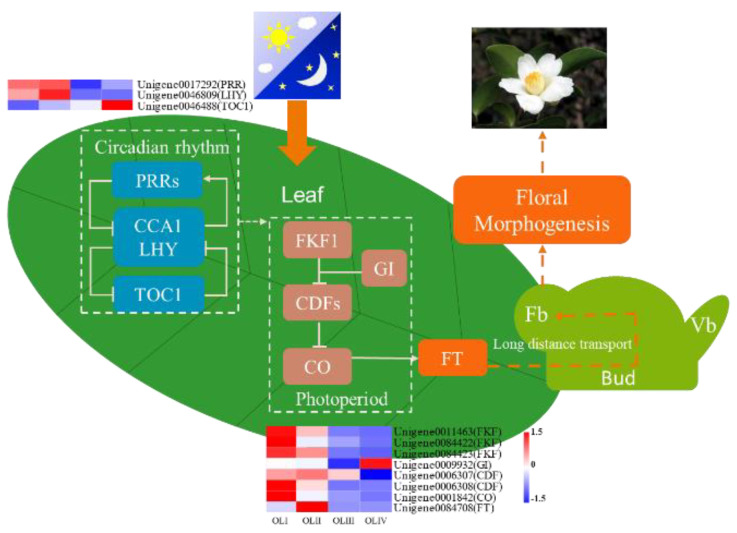
A diagram illustrating DEGs involved in signaling of circadian rhythm and photoperiod in old leaves to initiate flowering. Fb: Flower bud; Vb: Vegetative bud.

**Table 1 ijms-23-13021-t001:** Overview of the samples for RNA-seq.

Developmental Stages	Sample Name	Number of Clean Reads			Total_Mapped (%)		
		Rep 1	Rep 2	Rep 3	Rep 1	Rep 2	Rep 3
I	OLI	57,093,250	43,162,780	42,097,406	45,103,507 (79.00%)	34,259,949 (79.37%)	33,278,880 (79.05%)
II	OLII	52,057,704	46,463,330	38,605,766	41,838,924 (80.37%)	36,574,538 (78.72%)	30,836,344 (79.87%)
III	OLIII	48,287,926	43,057,968	41,933,040	37,256,384 (77.15%)	33,569,513 (77.96%)	32,828,511 (78.29%)
IV	OLIV	52,090,460	43,172,906	48,802,370	39,773,422 (76.35%)	32,573,906 (75.45%)	37,210,828 (76.25%)

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
