# Peer review of "Transcriptome Analysis Reveals Putative Induction of Floral Initiation by Old Leaves in Tea-Oil Tree (Camellia oleiferachanglin53’)"

_ijms, 2022, doi:10.3390/ijms232113021_

Round 1

Reviewer 1 Report

The manuscript is interesting. However, useful information is missing to understand clearly how the experiment was conducted and the data analyzed.

##Minor issues

- Lines 20-21, you said 5 genes and later on 55, you should rewrite for clarity

- Line 27, What is FT, you should define it

- Line 59, Do not start a sentence with an abbreviation, you should correct it throughout the manuscript

- Line 514: you add the internet link to OmicShare tools and when did you access it

##Major issues

- What about OLIII vs OLI, OLIV vs OLI,.... You should present results of these pairwise comparisons

- The tables are not well presented and are clouded! Add predicted in front of the gene name is not necessary.

- Sections 2.1 and 4.1: the morphological description and the supporting figure 1, and the sample collection are not clear. In all stages, leaves (a-d) are similar. Moreover, (e-h) are bud different developmental stages. Therefore, why were samples for transcritpome analysis collected from old leaves instead of buds that showed visual variations? You should explain your sampling strategy.

- Line 231-232, how a co-expression network of major enrichment pathways was constructed? You should provide details in the manuscript.

- 4.3 Section is not well described, you clearly explained how transcriptome assembly was performed and provide setting parameters.

- The overall materials and methods section is not well described and important information to understand how the experiment and analysis were conducted is missing.

- Section 4.7: You should present the qRT-PCR results in the main manuscript.

Reviewer 2 Report

1.     L134-143. The author did not explain the proportion of mapped reads per library ranged of each sample.

2.      L141-142. In this part, the author describes transcriptome sequencing with the De novo assembly of these clean reads. At present, the genome information of Camellia oleifera is very complete. Why didn't the author refer to known transcription?

3.     L231-232. You refer to "a co-expression network of major enrichment pathways was constructed ", however, the author did not mention the construction of co-expression network analysis in the materials and methods of the paper.

4.     L305. You refer to "However, no previous research in this area has been conducted in C. oleifera.", I suggest that the author should consult the literature further. At present, relevant research reports have appeared. Reference: doi.org/10.1186/s12870-022-03798-0

5.     L463-473. The author should specify the year of sample collection, so as to give readers a reliable reference.

6.     L163, 508. The expression of the screening indexes for significantly differentially expressed genes should be consistent.

7.     The material and methods section needs scientific sense and I would recommend the authors to review their corresponding parts. Add the number of biological replications for RNA extraction and library construction. (L520) details for genome version. (L525) add the reference for the method. The use of more than one housekeeping gene is recommended for validation purposes.

8.     You might want to follow the manuscript format of this journal. Including references, the format does not show any consistency. References 2, 16, 19, 21, 22, 25...

9.     Most of the references are too old. The author should quote more relevant research results from recent five years.

10.   L138. The Latin writing format of the full text should be consistent.

Round 2

Reviewer 1 Report

The section 4.2, lines 534-535:

- You should provide the Trinity version and setting parameters for the assembly

-  You should clearly explain how unigenes were obtained from the trinity assembly

-  You should provide parameters related to the quality assembly with trinity e.g Busco scores and mapping scores (back to the original assembly)

Round 3

Reviewer 1 Report

You did not answer my request to provide versions of all software used in this study in the manuscript. Trinity, DEseq2,...

Line 534, Provide Trinity assembly parameters

What is unigenes? How were they obtained from the trintity assembly?

Round 4

Reviewer 1 Report

Contigs from the trinity cannot be called unigenes since there are many duplicate contigs. To call them unigenes, you should perform additional analysis such as reducing duplicated contigs with CD-HIT-EST and clustering with TGICL tool. Here is an interesting discussion about unigene: https://www.researchgate.net/post/What_is_a_unigene

Therefore, I suggest replacing this term with an appropriate one.
